# Anoxygenic Photosynthesis in Photolithotrophic Sulfur Bacteria and Their Role in Detoxication of Hydrogen Sulfide

**DOI:** 10.3390/antiox10060829

**Published:** 2021-05-22

**Authors:** Ivan Kushkevych, Veronika Bosáková, Monika Vítězová, Simon K.-M. R. Rittmann

**Affiliations:** 1Department of Experimental Biology, Faculty of Science, Masaryk University, 62500 Brno, Czech Republic; 451645@mail.muni.cz (V.B.); vitezova@sci.muni.cz (M.V.); 2Department of Biology, Faculty of Medicine, Masaryk University, 62500 Brno, Czech Republic; 3Archaea Physiology & Biotechnology Group, Department of Functional and Evolutionary Ecology, Universität Wien, 1090 Vienna, Austria

**Keywords:** hydrogen sulfide, toxicity, waste water treatment, anoxygenic photosynthesis

## Abstract

Hydrogen sulfide is a toxic compound that can affect various groups of water microorganisms. Photolithotrophic sulfur bacteria including *Chromatiaceae* and *Chlorobiaceae* are able to convert inorganic substrate (hydrogen sulfide and carbon dioxide) into organic matter deriving energy from photosynthesis. This process takes place in the absence of molecular oxygen and is referred to as anoxygenic photosynthesis, in which exogenous electron donors are needed. These donors may be reduced sulfur compounds such as hydrogen sulfide. This paper deals with the description of this metabolic process, representatives of the above-mentioned families, and discusses the possibility using anoxygenic phototrophic microorganisms for the detoxification of toxic hydrogen sulfide. Moreover, their general characteristics, morphology, metabolism, and taxonomy are described as well as the conditions for isolation and cultivation of these microorganisms will be presented.

## 1. Introduction

Hydrogen sulfide is a colorless gas with a characteristic odor that is soluble in various liquids including water. In nature, hydrogen sulfide is an intermediate of the sulfur cycle. The main producers of hydrogen sulfide in the environment are sulfate-reducing microorganisms [1,2,3,4,5,6]. Hydrogen sulfide occurs in volcanic gases and it is produced by some microbial metabolic processes during decomposition of plant and animal proteins. Hydrogen sulfide is highly toxic, and even in small doses can cause fatal poisoning. It inhibits the enzyme cytochrome *c* oxidase and thus prevents tissues from using molecular oxygen (O_2_) [7,8]. This is mainly manifested in the central nervous system by paralysis of the respiratory center. In some cases, it may accumulate in aquatic environments. However, this accumulation can have fatal consequences for such ecosystems. As a result of hydrogen sulfide poisoning, mass deaths of aquatic animals and fish occur. Furthermore, the decomposition of these organisms leads to a further increase in the amount of hydrogen sulfide. Another problem with hydrogen sulfide-polluted water is that it is very difficult and very expensive to detoxify contaminated water [1]. Moreover, in most cases, it is practically impossible to remove all of hydrogen sulfide with currently existing methods.

Photolitotrophic sulfur bacteria are a group of microorganisms that includes the families *Chlorobiaceae* (green sulfur bacteria) and *Chromatiaceae* (purple sulfur bacteria) [9]. These names refer to their coloration, which are due to the different content of photosynthetic pigments, such as bacteriochlorophylls and carotenoids [10]. These specialized bacteria have developed a special form of photosynthesis, the so-called anoxygenic photosynthesis, which takes place under anaerobic conditions and oxidizes reduced sulfur compounds as an electron donor [11]. The process of anoxygenic photosynthesis thus differs from oxygenic photosynthesis, which is characteristic of cyanobacteria and plants. Green sulfur bacteria (GSB) and purple sulfur bacteria (PSB) of the above families may metabolize hydrogen sulfide. High concentrations of this compound often occur in aqueous layers of molecular sulfur-rich sediment, which is reduced by two groups of microbial communities: sulfate-reducing and sulfur-reducing microorganisms. The sulfate or sulfur is being used as a terminal electron acceptor in anaerobic respiration. On the one hand, sulfur anoxygenic phototrophs oxidize sulfur compounds to sulfates and molecular sulfur, and on the other hand, these compounds are reduced. Therefore, the described microorganisms are in close interaction with one another and are involved in the sulfur cycle in water layers and occur in nature in general [12].

Although *Chlorobiaceae* and *Chromatiaceae* have been known for almost a century, they have not been thoroughly studied, as very few researchers have dealt with this group of microorganisms [10,11,13]. However, these interesting bacteria could be used in the biotechnology industry due to their ability to utilize hydrogen sulfide. 

This work aims to summarize the recent knowledge on the above-mentioned families of bacteria with emphasis on the description of the mechanism of anoxygenic photosynthesis, their cultivation, and an outline of their possible use in industry.

## 2. General Characteristics of Photoautotrophic Bacteria

The most remarkable and at the same time common feature of all GSB and PSB are the ability of anoxygenic photosynthesis based on bacteriochlorophyll-mediated processes [10]. Different anoxygenic phototrophic bacteria contain several types of bacteriochlorophylls and various carotenoids as pigments, which function to transform light into chemical energy and give cultures a strong coloration that differs in pigment content from different shades of green, yellow-green, brown-green, brown, red, pink, purple, up to blue [14]. Photosynthesis in phototrophic sulfur bacteria depends on the O_2_ content of the environment because the synthesis of their photosynthetic dyes is suppressed by O_2_. Unlike cyanobacteria and eukaryotic algae, phototrophic sulfur bacteria are unable to use water as an electron donor and do not produce O_2_. They use only one photosystem and require electron donors with a lower redox potential than water. Sulfur and its reduced compounds are most often used as donors, but also hydrogen and many other small organic molecules [9].

Green sulfur bacteria form a phylogenetically consistent and isolated group of bacteria. They differ in that inside their cells there are special light-harvesting complexes called chlorosomes, which contain bacteriochlorophylls and carotenoids. GSB also differ from other phototrophic organisms in the chemical structure of bacteriochlorophyll antennas. The same antenna composition was found only in the phylogenetically distant family *Chloroflexaceae* [13]. Chlorosomes of GSB contain bacteriochlorophyll (BChl) c, d, or e. Green-colored bacteria contain mainly BChl c or d; others are colored orange or brown, due to the high content of carotenoids [15].

Unlike GSB, PSB do not contain chlorosomes and their photosynthetic apparatus, including pigments, is stored in one or more of the extended intracellular systems of the cytoplasmic membrane. These systems consist of folds, tubules, vesicles, or lamellae. The most common photosynthetic dye found in PSB is bacteriochlorophyll a or b. *Chromatiaceae* also contains a large number of auxiliary dyes, such as spirilloxynthine, rhodopine, or okenone carotenoids [16].

Due to their limited physiological flexibility, the ecological niche of GSB is rather narrow. All known species are typically aquatic microorganisms and inhabit illuminated anoxygenic layers of lakes or littoral sediments. In some of these ecosystems GSB play a major role in the transformation of carbon and sulfur compounds. Another phenotypic feature of ecological significance is the adaptation to very low light intensities. Compared to other phototrophic microorganisms, GSB are able to inhabit the lowest parts or sediments of ecosystems. The cells of most species belong morphologically to the most inconspicuous members of natural bacterial communities. An exception are phototrophic consortia, which are permanent associations of GSB with chemolithotrophic bacteria. At present, these phototrophic consortia represent one of the most developed symbioses in the prokaryotic world [13].

Purple sulfur bacteria inhabit the same sites as green sulfur bacteria, and some may even live in a symbiosis-like relationships with them. In general, however, *Chromatiaceae* live above *Chlorobiaceae* because they need higher light intensity and lower hydrogen sulfide concentrations for photosynthesis [10].

Both GSB and PSB are of the gram-negative type. All GSB described so far are rods and their size is around 1 μm. PSB are more diverse in shape; the shape of cells varies from species to species, from cocci, through rods, to spirals. However, it can also vary during the cell life cycle depending on external conditions. PSB are slightly larger than GSB and reach a size of up to 3 μm. Most types of PSB are motile with one or more flagella, while GSB lack a flagellum and are therefore immobile. Some GSB species, such as *Chlorobium limicola*, form streptococcal-like chains. In addition, these chains may, depending on the culture conditions, be coated with a mucous layer. PSB occur either singly or in pairs. For later oxidation, the PSB store sulfur inside the cell in so-called granules, and the GSB store it outside the cell, where they are held at the membrane by special mechanisms. Some PSB species have gas sacs inside the cell that allow them to float in a low-density environment [17].

## 3. Anoxygenic Photosynthesis

The conversion of light into chemical energy is an essential process for life, and photosynthetic reaction centers play a major role in this mechanism. These are special complexes of proteins and chlorophylls in the nucleus of the photosynthetic system, which are excited after irradiation and absorption, thus releasing energy that allows electrons to pass through the photosynthetic membrane. Two types of reaction centers are known. The first type is photosystem 1 (PS1), which is found in chloroplasts of cyanobacteria and GSB. The second type is photosystem 2 (PS2), which is found in chloroplasts and cyanobacteria. This type of reaction center is also found in PSB [18].

### 3.1. The Photosystem of the Family Chlorobiaceae

In the case of GSB, the antenna complex is found in special formations called chlorosomes. These chlorosomes are located on the inner side of the inner cell membrane and perform the function of absorbing light radiation and transferring energy to the photosynthetic reaction center. They differ from all known photosynthetic antenna complexes due to their pigment–pigment arrangement, instead of the typical pigment–protein, such as the *Chromatiaceae*. Chlorosomes contain lipids, small amounts of protein, carotenoids (chlorobactein, neurosporein, and lycopene) and bacteriochlorophylls. The function of proteins in chlorosomes has not been elucidated, but they are thought to ensure the stability of bacteriochlorophylls and maintain the stable ovoid shape of chlorosomes. In addition to chlorosomes, GSB contain another unique antenna complex, called the Fenna–Matthews–Olson (FMO) protein. This protein transports electrons from chlorosomes to the reaction center. Most proteins containing bound bacteriochlorophylls are insoluble in water, with the exception of FMO protein [19].

In the case of GSB, the energy of light radiation is transmitted to the reaction center by means of chlorosomes. These structures described above capture high radiation efficiency, thanks to the huge amount of bacteriochlorophylls contained (about 200,000 molecules per chlorosome). The reaction center contains an average of 500 bacteriochlorophyll molecules, and one chlorosome binds to up to forty such reaction centers. Bacteriochlorophylls are arranged in chlorosomes in tubules with an absorption maximum between 720–750 nm. The energy transfer of light radiation passes through these tubules to the so-called baseplate, formed by bacteriochlorophyll, and from there the energy passes into the FMO protein (absorption maximum FMO 808 nm). Via FMO protein, energy is already transferred to the reaction center of the photosynthetic apparatus of *Chlorobiaceae* [20].

### 3.2. The Photosystem of the Family Chromatiaceae

The antenna system in PSB consists of two main types of light-harvesting complexes, LH1 and LH2. The LH1 complex is found in all species of this family and surrounds the reaction center and forms with it the so-called core of the photosynthetic complex. The LH2 complex is located in the periphery of this nucleus and does not occur in some species. Both LH1 and LH2 are large oligomers composed of heterodimers of transmembrane polypeptides (α and β) associated with bacteriochlorophylls and carotenoids. Although both subunits are structurally almost identical, the LH1 complex absorbs light radiation of a longer wavelength (870–960 nm) than the LH2 complex (800–850 nm).

The LH1 complex is evenly distributed around the reaction center and forms a closed and slightly elliptical cylinder composed of 16 pairs of helical αβ-polypeptides, 32 bacteriochlorophyll a, 16 carotenoids (spirilloxanthin) and 16 Ca^2+^ ions. The ratio between the content of pigments and Ca^2+^ ions is stoichiometrically constant. As with LH2, α-polypeptides are found inside LH1 and β-polypeptides outside the ring [21].The LH2 complex is a typical membrane protein of cylindrical structure, containing 27 bacteriochlorophylls and 9 carotenoids. It contains 9 αβ-heterodimers that form a circular aggregate [22].

## 4. Pigments

Bacteriochlorophylls are the major photopigments of photosynthetic bacteria. They serve as light-harvesting pigments in the antennas, but are also located in the reaction center. Chemically, all bacteriochlorophylls are derivatives of porphyrin, which forms an isocyclic ring with magnesium in the center and a propionate ester group at C17. Bacteriochlorophylls ***a*** and ***b*** are the most common photopigments in photosynthetic bacteria. Bacteriochlorophyll a occurs in most PSB as the only dye for antennas and reaction centers. In GSB, this pigment occurs only in the reaction center. Carotenoids occur in GSB and PSB as accompanying auxiliary dyes. These include, for example, dyes (1) spirilloxanthine lines such as lycopene, rhodopine, or spirilloxanthin, (2) rhodopine lines such as lycopene, lycopenal, lycopenol, rhodopine, (3) alternative spirilloxanthine lines such as spheroidine, (4) okenone lines is an okenon [23].

The regulation of the formation of photosynthetic pigments in photolitotrophic sulfur bacteria can be influenced by external conditions, such as light intensity or the presence of O_2_. Like all other phototrophic organisms, phototrophic bacteria are able to regulate bacteriochlorophyll and carotenoid content depending on light intensity. Based on the knowledge that cell growth is also a function of light intensity, the dependence of the concentration of pigments C on the light intensity I was measured and described by a simple relation *C* = *K* + (*A*/l), where *K* and *A* are empirical constants [24]. Because bacteriochlorophylls incorporate into the intracytoplasmic membrane system, there are three possible ways in which bacteriochlorophyll content could be affected: by changing the specific pigment content in membranes, by changing the amount of photosynthetic membrane systems with constant specific pigment content, and a combination of both.

## 5. Sulfur Metabolism

Phototrophic sulfur bacteria are characterized by the ability to oxidize various inorganic sulfur compounds. Such compounds include, for example, hydrogen sulfide, elemental sulfur, and thiosulfate. Hydrogen sulfide oxidation can be mediated by several enzymes, including flavocytochrome c sulfide dehydrogenase, sulfide: quinone oxidoreductase, and the Sox enzyme system.

**Flavocytochrome c**. Flavocytochrome c is a soluble, periplasmic enzyme composed of large, sulfur-binding FccB flavoprotein subunits and smaller FccA cytochrome c subunits. In vitro, flavocytochrome effectively catalyzes the transfer of electrons from hydrogen sulfide to small cytochromes, and these electrons can be further used in photosynthetic reaction centers. However, in vivo, the role of flavocytochrome is still unclear for several reasons. Such reasons include, for example, that some species of GSB and PSB oxidize hydrogen sulfide but do not synthesize the enzyme. This is due to the fact that both families have an alternative oxidation pathway and that is sulfide:quinone oxidoreductase [14,15].

**Sulfide:quinone oxidoreductase**. Sulfide:quinone oxidoreductase (SQR) catalyzing the oxidation of hydrogen sulfide by isoprenoid quinones has been found not only in chemotrophic and phototrophic prokaryotes, but also in some mitochondria [25,26]. Membrane-bound SQR activity has been biochemically demonstrated in GSB and PSB and probably provides electrons to photosynthetic electron flux using a complex of quinone-oxidizing Rieske FeS protein and cytochrome *b* [11,27]. The genomic sequences of all strains of GSB encode one or two homologues of SQR, including *Chlorobium ferroxidans*, which cannot utilize sulfur as the sole electron donor [28]. This organism may benefit from SQR activity in addition to its energetic metabolism. It could also use SQR as a protective mechanism to remove sulfide, which inhibits growth when present in high concentrations. SQR homologues in GSB are flavoproteins with a putative mass of approximately 53 kDa and each contain all three conserved cysteine residues that are essential for sulfide oxidation [29].

**Sox enzymatic system**. In *Rhodovulum sulfidophilum*, a species of purple non-sulfur bacteria of the family *Rhodobacteraceae*, the Sox system catalyzes the oxidation of thiosulfate to sulfate [30]. However, in *Allochromatium vinosum* mutants lacking flavocytochrome c, sox genes, or both, sulfate oxidation proceeds at the same rate as wild-type. This indicates that SQR plays a major role in sulfate oxidation. The occurrence of sox genes in GSB and PSB suggests a correlation with the ability to oxidize thiosulfate [11].

**Oxidation of polysulfides**. In some species of green and PSB, the primary product of sulfide oxidation are polysulfides. The use of externally added polysulphides was studied in *Allochromatium vinosum* and *Thiocapsa roseopersicina*. Both organisms have readily used these compounds as electron donors [31,32]. It is currently unknown how polysulfides are converted to sulfur globules. Theoretically, this process could occur purely chemically because polysulfides are in equilibrium with elemental sulfur [31].

**Sulfur absorption**. Most GSB and PSB are able to oxidize externally supplied solid, virtually insoluble elemental sulfur. This is a very demanding biochemical process in that elemental sulfur is extremely hydrophobic and practically insoluble in water. In addition, elemental sulfur cannot be attacked by oxidation reactions in the absence of O_2_ under anoxic conditions. In PSB, sulfur is first taken up by cells to form intracellular sulfur globules, and then oxidized in them [33,34,35]. Unlike PSB, GSB do not form intracellular globules. Very little is known about the absorption and transformation of elemental sulfur in phototrophic sulfur bacteria. Enzymes catalyzing the absorption and oxidation of externally added elemental sulfur have not yet been isolated from phototrophs. The use of solid elemental sulfur must include binding, sulfur activation, and intracellular transport.

## 6. Taxonomy

The traditional division of species, genera, and even families of phototrophic bacteria was previously based on a number of morphological properties, such as cell shape and size, flagellar or intracellular membrane structure formation, the representation and amount of photosynthetic pigments, and physiological properties such as the ability to utilize nitrogen and carbon from various substrates [9,36]. Later, using protein and nucleic acid sequencing techniques, molecular data were available and used to analyze the phylogenetic relatedness of bacterial strains. Due to the availability of methods, proteins were the first macromolecules used to determine the relationship of different species of bacteria. Examples of such proteins are cytochromes type c and ferredoxins. Based on the primary sequence and tertiary structure of cytochromes type c, the first phylogenetic tree of PSB was constructed [37].

Recently, the 16S rRNA sequencing method has been widely used to assemble phylogenetic trees. This molecule is universally distributed among prokaryotic organisms and is considered phylogenetically conserved. Previously, the method was based on a mere comparison of oligonucleotide catalogs obtained by cleavage of a molecule with T1 RNase [38]. With the development of techniques, complete sequencing of 16S rRNA has occurred, so the entire sequence of this molecule is now available for comparison. Although this method is superior to all previous attempts to establish phylogenetic relationships, it is only an assessment of a single molecule from the whole bacterium. Therefore, a more suitable approach would be to use concatenated ribosomal protein trees, whole genome comparison and/or to combine the result obtained with different approaches, supported by sequencing and comparison of 16S rRNA.

### 6.1. Family Chlorobiaceae

Green sulfur bacteria form a phylogenetically very separated group of bacteria (<83% 16S rRNA genetic similarity) and form a unique taxonomic group (88–100% 16S rRNA similarity) in the class *Chlorobia*. All species are divided into several genera and one family *Chlorobiaceae* [19]. The phylogenetic relationships of a large number of GSB were determined using the 16S rRNA and fmoA (gene for Fenna–Matthews–Olson protein) gene sequences [14]. Fenna–Matthews–Olson protein is a water-soluble complex that was analyzed as the first pigment–protein complex by X-ray spectrometry [39,40].

Phylogenetically, GSB form four main groups, which currently represent the described genera *Chlorobium*, *Prosthecochloris*, *Chlorobaculum,* and *Chloroherpeton*. A typical example of the genus *Chlorobium* is the species *Chlorobium limicola* [41]. Other species include, for example, *Chlorobium phaeobacteroides, Chlorobium clathratiforme*, formerly called *Pelodictyon clathratiforme, Chlorobium ferrooxidans,* and *Chlorobium phaeovibrioides.* A typical example of the genus *Prosthecochloris* is the species *Prosthecochloris aestuarii.* Other species include, for example, *Prosthecochloris vibrioformis*, formerly called *Chlorobium vibrioforme* [42].

A typical example of the genus Chlorobaculum is the species Chlorobaculum tepidum, formerly called Chlorobium tepidum. Other species include, for example, Chlorobaculum limnaeum, Chlorobaculum thiosulfatiphilum, and Chlorobaculum parvum [40]. 

### 6.2. Family Chromatiaceae

Previously, the family *Chromatiaceae* was described to include the genus *Ectothiorhodospira*, dandelion sulfur bacteria that store sulfur outside the cell [43]. With the establishment of the family *Ectothiorhodospiraceae*, the family *Chromatiaceae* was introduced as exclusively containing phototrophic sulfur bacteria capable of forming sulfur globules inside the cell [9]. Over the last few years, 16S rRNA sequencing has been completed for most *Chromatiaceae* species. Based on the analysis of 16S rRNA and some morphological features, the current phylogenetic tree of the *Chromatiaceae* family was compiled, which consists of three main groups. The first are true marine and halophilic species; the second are species that are motile due to the polar flagellum and do not contain gas sacs and are primarily freshwater; the third group are freshwater immobile species forming gas sacs [10].

## 7. Isolation and Cultivation

Since the first experimental studies of PSB conducted by Winogradsky in 1888, the growth of these bacteria in mixed enriched cultures has become commonplace. These well-known Winogradsky columns are housed in tall glass cylinders with plant debris, CaSO_4_, anaerobic mud, and natural water. In this way, they are incubated in partial shade on the north window. Today, however, synthetic media are described that provide isolation and selectivity for certain species of green and purple bacteria. Samples from freshwater, such as sulfur springs, lakes, rivers or puddles, soil and mud samples, or marine sediment, can be used to isolate green and purple sulfur bacteria. During the summer and autumn, when the leaves fall and are decomposed by sulfate-reducing bacteria, the concentration of hydrogen sulfide in the environment increases and thus the population of photolitrophic and photochemoorganotrophic sulfur bacteria increases, which subsequently produces observable coloration. This period is the most suitable for sampling. By culturing the samples in a medium containing hydrogen sulfide and inorganic salts, mixed cultures are formed, which can turn red to green depending on the predominance of the species in the sample. However, *Chlorobiaceae* and *Chromatiaceae* are occur in mixed cultures. Selective conditions are used to isolate species from only one family. These conditions include adjusting the culture medium, lighting, or temperature. After observing the enrichment, sooner or later the predominance of one of the families will show a strong color.

### 7.1. Selective Media

For some species, the concentration of hydrogen sulfide in the medium is critical and most species of both families are inhibited by a high concentration (concentrations above 0.1% hydrogen sulfide in the medium). In general, however, bacteria of the *Chlorobiaceae* family are able to tolerate higher concentrations. Therefore, one of the possible ways to ensure the selectivity of the medium is to increase or decrease the concentration of added hydrogen sulfide [44]. Another way to selectively isolate only one of the families is to adjust the pH of the medium. Species from the family *Chlorobiaceae* require a more acidic environment, and therefore it is necessary to adjust the pH of the solution to a value between 6.6–6.9. For isolation of species from the family *Chromatiaceae* it is necessary to have a higher pH in the range from 7.2 to 7.4.

### 7.2. Selection Using Physical Conditions

Cultivation conditions that can be varied in the laboratory include the intensity and wavelength of the light and the ambient temperature at the cultivation site. The ambient temperature can be changed and maintained by cultivating in thermostats. The photolitotrophic sulfur bacteria described above are, with a few exceptions, mesophilic, but species of the *Chlorobiaceae* family require a slightly higher temperature (about 35 °C) than *Chromatiaceae* (about 30 °C). However, temperature is not a very suitable selective condition and is only used to isolate species that are thermophilic, for example from the family *Chlorobiaceae* species *Chlorobium tepidum*.

In contrast to temperature, light intensity is a more suitable selective condition. Due to the fact that the family *Chromatiaceae* is located in the natural environment above the layer of bacteria of the family *Chlorobiaceae*, and thus creates a natural screen for light radiation, and therefore GSB do not tolerate high light intensities. In addition, thanks to their powerful light-collecting antennas, they are able to effectively use radiation of very low intensity. Cultivation under very high light intensity (1000–2000 lx) is selective for small, fast-growing species of the *Chromatiaceae* family, such as *Thiocapsa roseopersicina*. The light intensity in the range of 50–300 lx is selective for large flagellate species of the *Chromatiaceae* family, for example *Chromatium oknoii*. Selective for the family *Chlorobiaceae* is the intensity between 5–50 lx.

Using the knowledge that different photosynthetic dyes achieve maximum absorption at other wavelengths, the wavelength of light can be used as a selection method. The easiest way to achieve cultivation below a certain wavelength in the laboratory is to use light filters. These filters transmit radiation only at a certain wavelength and absorb the others. Dyes, which occur in the family *Chlorobiaceae*, absorb most in the visible region in the blue part (520–430 nm) and in the near UV (300–200 nm). In the region of red visible light (650–750 nm) and near IR (800–900 nm), they achieve the maximum absorption of the dye, which is found in the family *Chromatiaceae*.

## 8. Application of Anoxygenic Phototrophs in Hydrogen Sulfide Detoxication

Due to their unusual metabolism, phololitotrophic sulfur bacteria have great potential in the use of biotechnological processes (Figure 1). Such processes include, for example, the production of feed for cattle and fish. Some species of *Chlorobiaceae* (such as *Chlorobium limicola*) use glycogen as a storage substance, which they can produce with light from simple inorganic salts and hydrogen sulfide. Glycogen is a branched polysaccharide, the decomposition of which in the body yields energy and glucose. At present, glycogen is obtained by laborious and particularly expensive methods, for example from pig liver. Therefore, it is obvious to obtain another source of this sugar, which could then be used as a cheap and energy-rich feed for farm animals. This source could be the cultivation and subsequent separation of photolitotrophic GSB containing glycogen. If the right cultivation conditions were designed and optimized, this method might be technically simple and very cheap. Application of these bacteria in industry might be the use of their ability to survive in the environment with hydrogen sulfide and the ability to remove it from this environment [45]. Using GSB and PSB to purify hydrogen sulfide-contaminated water might be very cheap and technically simple.

The scheme presented in Figure 1 shows a possible method to purify biogas from hydrogen sulfide. The purification process with the feed of raw biogas (Figure 1, step 1) through a filter, which ensures even dispersion of gas droplets in the first vessel (Figure 1, step 2). It contains a suitable species of GSB that can survive in a high concentration of hydrogen sulfide while reducing its concentration through metabolic processes. The gas from the first vessel is directed to the second vessel (Figure 1, step 3) in the same way as the raw biogas through the filter. This vessel contains a mixed culture of GSB and PSB, which require a lower concentration of hydrogen sulfide. From the second vessel, biogas exits that contains a low hydrogen sulfide content. It is removed in a third vessel (Figure 1, step 4) of the species PSB, which is able to utilize very low hydrogen sulfide concentration. The purified biogas (Figure 1, step 5) may then be harvested.

Another possible use of these bacteria could be the purification of biogas from anerobic digestion plants and thus increase its quality. Biogas is one of the possible future renewable energy sources. It is produced by anaerobic consortia of organisms by the decomposition of organic material [46]. Such material can be, for example, livestock manure in biogas plants. The raw biogas contains a mixture of gases such as methane, carbon dioxide, hydrogen, and hydrogen sulfide. The aim is to obtain pure methane, which can then be used to produce heat, electricity, or to drive vehicles. At present, there is no effective biological method that can completely separate the gas from the impurities; only physico-chemical methods are used, which are very demanding and expensive. Unlike pure methane, the use of polluted gas is inefficient and it is not possible to use all the potential energy of this gas [47]. For efficient purification, it would be possible to use the ability of photolitotrophic sulfur bacteria to reduce hydrogen sulfide to atomic sulfur, which is very easy to separate from gases. In the process of anoxygenic photosynthesis, they can also use molecular hydrogen as an electron donor, and carbon dioxide gas is used in building organic matter. Thus, it is theoretically possible to obtain almost pure methane from raw biogas using these microorganisms. If this method would work, it could become the first effective method for biogas purification. At the same time, it would be technically undemanding and inexpensive.

## 9. Conclusions

The general characteristics of photolitotrophic sulfur bacteria from the family *Chlorobiaceae* and *Chromatiaceae* were described in this work, which differ in their morphology, photosynthetic apparatus, and mechanism of anoxygenic photosynthesis. However, a common property of both families is the same electron donor and thus reduced sulfur compounds, such as hydrogen sulfide. In the process of anoxygenic photosynthesis, they oxidize to atomic sulfur, which is stored in the form of globules in the *Chromatiaceae* family inside the cell and in the *Chlorobiaceae* family outside the cell.

The metabolism of anoxygenic photosynthesis is an important process for the acquisition of metabolic energy in both families. It also plays an important role in the natural sulfur cycle in the environment. The process involves the absorption of light radiation, which occurs in light-collecting apparatuses. These apparatuses differ in GSB and PSB. In GSB, photosynthetic apparatus is found in special formations called chlorosomes; in *Chromatiaceae* it is found in intracellular membrane folds. The absorption of light radiation is mediated by pigments, which are bacteriochlorophylls and accompanying dye carotenoids. The process of anoxygenic photosynthesis further involves the oxidation of hydrogen sulfide to atomic sulfur. Oxidation is provided by several metabolic pathways, which are currently not fully elucidated. The taxonomy of photolitotrophic bacteria has undergone significant changes, previously based only on a number of morphological and physiological features, such as cell shape and size, or the ability to utilize atmospheric nitrogen. With the later development of sequencing techniques, the taxonomic distribution of both families was adjusted based on a comparison of the sequence of 16S rRNA and several genes specific for both families.

Due to their metabolism, photolitotrophic sulfur bacteria could be used in the biotechnology industry. Such uses include, for example, the production of biomacromolecules such as glycogen. Glycogen is a branched polysaccharide, the decomposition of which in the body results in a large gain of energy and glucose. However, its production is now demanding and expensive. Another possible use is the purification of polluted waters with hydrogen sulfide, which occurs with increased decomposition of organic matter by sulfate-reducing bacteria. Photolitotrophic sulfur bacteria could also be used to purify raw biogas. This gas is produced by anaerobic decomposition of organic matter in biogas plants and seems to be one of the possible renewable sources. However, in addition to the necessary methane, it also contains undesirable impurities, among which belongs hydrogen sulfide. There is currently no effective method to separate pure methane from impurities.

Although these bacteria have been known for almost a century, their physiological and biotechnological potential has not yet been fully studied. Some metabolic processes are still unclear, including metabolism of hydrogen sulfide oxidation or sulfur uptake by the cell. Further research on biochemistry, structural and molecular biology, and biotechnology is needed to fully understand their metabolic capability.

## Figures and Tables

**Figure 1 antioxidants-10-00829-f001:**
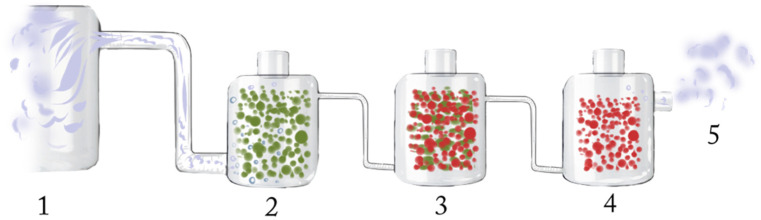
The scheme of the purification of biogas. (1) raw biogas, (2) green sulfur bacteria, (3) mixture of green and purple sulfur bacteria, (4) purple sulfur bacteria, (5) pure biogas.

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
