# Peer review of "Anoxygenic Photosynthesis in Photolithotrophic Sulfur Bacteria and Their Role in Detoxication of Hydrogen Sulfide"

_antioxidants, 2021, doi:10.3390/antiox10060829_

Round 1

Reviewer 1 Report

I wish first to underline that I am not an expert in the field covered by this review article although I have a broad knowledge of microbiology and redox biology. Therefore I shall be unable to point out mis quotations or lack of singificant contributions if there is any. Hence my comment should be taken as coming from an average reader willing to step out of his usual field and get a flavour of an unfamiliar one. Under such criteria I found the review of great interest, well written and have the impression I learned and understood new facettes of microbiology, evolution and redox biology from reading it.

Author Response

Dear Reviewer,

We would like to thank you for your time reviewing our manuscript. The text was carefully revised and improved our work. We have corrected the manuscript according to the recommendations all reviewers.

Thank you for your time, positive evaluation of our work and your help.

Best wishes,

The authors

Reviewer 2 Report

The review presents an extensive information on the photosynthetic sulfur bacteria (including metabolism, physiology, cultivation). In the same time, according to the title of the manuscript, one of the main goals of the work was to characterize possibility of PSB application in biotechnology. In fact, this section (8) contains only one reference and does not provide complete data on this issue. Thus, the section 8. Application of....... should be improved and the data on the current state of the problem and apropriate references should be included in the review.

Author Response

Dear Reviewer,

we would like to thank you for your time reviewing our manuscript. The text was carefully revised and we are thankful for your important and constructively critical comments, which improved our work. We have corrected the manuscript according to your recommendations and written our responses below

Thank you for your time, positive evaluation of our work and your help.

Best wishes,

The authors

The review presents an extensive information on the photosynthetic sulfur bacteria (including metabolism, physiology, cultivation). In the same time, according to the title of the manuscript, one of the main goals of the work was to characterize possibility of PSB application in biotechnology. In fact, this section (8) contains only one reference and does not provide complete data on this issue. Thus, the section 8. Application of....... should be improved and the data on the current state of the problem and apropriate references should be included in the review.

It is true that section 8 contains only one reference. This section was written based on analyzing and summarizing all literature data and include analyze and recommendation by the authors for application of anoxygenic phototrophs in hydrogen sulfide detoxication. It is not literature data where we can cite references, it is our own thoughts and recommendations to possibly set-up suh a detoxification system.

Reviewer 3 Report

The authors have reviewed aspects of anoxygenic photosynthesis by photolithotrophic sulfur bacteria in particular their use of hydrogen sulfide as an electron source. In general, the review is interesting and informative. There were minor issues which are described below. The nomenclature is sometimes confusing, as several synonyms are used to name the bacteria. The writing style is sometimes awkward. The manuscript would benefit from editing.

  1. Line 29 : « small doses ».
  2. Line 31, references 7 and 8 : do these address the point being made? Are self-citations necessary here?
  3. Line 36: “Another problem with hydrogen sulfide pollution”.
  4. Reference 9: a title is missing. Same in references 21, 33.
  5. Line 44: “reduces reduced”. Please re-word. Do you mean oxidizes?
  6. Line 46: “oxygenic”.
  7. Line 48: “High concentrations of this compound …, which is reduced by sulfate and sulfur-reducing microorganisms”. Do you mean that it is produced by these microorganisms? What is the metabolic basis for this production? Is sulfate or sulfur being used as a terminal electron acceptor in anaerobic respiration? This should be explained.
  8. Line 57: “group of (micro)organisms” is repeated twice on this line.
  9. Figure 1: “What is this review about?”. This figure is not necessary. The structure of the article can be read from the subtitles. The space from this figure could be used to illustrate some other aspects, such as photosynthetic centers and pigments, sulfur oxidation or phylogeny.
  10. Line 67: “is”.
  11. Sentence at line 86 should be attached to previous paragraph.
  12. Line 167, remove “from”.
  13. Line 176, “are”.
  14. Section on flavocytochrome c, line 204, a reference should be added.
  15. Line 226. Do not refer to sulfate oxidation in this section. Sulfate cannot be oxidized further.
  16. Under section 6.2, Family Chromatiaceae, add a reference.
  17. Figure 2. More details should be provided in then text. Explain the particular succession of sulfur bacteria depicted in Figure 2.
  18. Reference style is inconsistent, for example journal abbreviations.

Author Response

Dear Reviewer,

we would like to thank you for your time reviewing our manuscript. The text was carefully revised and we are thankful for your important and constructively critical comments, which improved our work. We have corrected the manuscript according to your recommendations and written our responses below.

Thank you for your time, positive evaluation of our work and your help.

Best wishes,

The authors

The authors have reviewed aspects of anoxygenic photosynthesis by photolithotrophic sulfur bacteria in particular their use of hydrogen sulfide as an electron source. In general, the review is interesting and informative. There were minor issues which are described below. The nomenclature is sometimes confusing, as several synonyms are used to name the bacteria. The writing style is sometimes awkward. The manuscript would benefit from editing.

    1.  

Line 29 : « small doses ».

It was corrected.

    1.  

Line 31, references 7 and 8 : do these address the point being made? Are self-citations necessary here?

These references are necessary because there are indicated information about hydrogen sulfide inhibition of enzyme cytochrome c oxidase and thus prevents tissues from using oxygen.

    1.  

Line 36: “Another problem with hydrogen sulfide pollution”.

It was corrected.

    1.  

Reference 9: a title is missing. Same in references 21, 33.

    1.  

It was corrected.

    1.  

Line 44: “reduces reduced”. Please re-word. Do you mean oxidizes?

Yes, oxidizes. It was corrected. Thank you very much.

    1.  

Line 46: “oxygenic”.

It was corrected.

    1.  

Line 48: “High concentrations of this compound …, which is reduced by sulfate and sulfur-reducing microorganisms”. Do you mean that it is produced by these microorganisms? What is the metabolic basis for this production? Is sulfate or sulfur being used as a terminal electron acceptor in anaerobic respiration? This should be explained.

Yes, I meant these microorganisms (sulfate-reducing and sulfur-reducing microorganisms). It was corrected and explained. Thank you.

    1.  

Line 57: “group of (micro)organisms” is repeated twice on this line.

It was corrected.

    1.  

Figure 1: “What is this review about?”. This figure is not necessary. The structure of the article can be read from the subtitles. The space from this figure could be used to illustrate some other aspects, such as photosynthetic centers and pigments, sulfur oxidation or phylogeny.

This figure has been deleted from manuscript.

    1.  

Line 67: “is”.

if reviewer mean: “The most remarkable and at the same time common feature of all GSB (green sulfur bacteria) and PSB are the ability of anoxygenic photosynthesis….”, there should be “are”.

    1.  

Sentence at line 86 should be attached to previous paragraph.

    1.  

It was corrected.

    1.  

Line 167, remove “from”.

from” has been deleted.

    1.  

Line 176, “are”.

It was corrected.

    1.  

Section on flavocytochrome c, line 204, a reference should be added.

The references have been added.

    1.  

Line 226. Do not refer to sulfate oxidation in this section. Sulfate cannot be oxidized further.

Thank you. It was corrected.

    1.  

Under section 6.2, Family Chromatiaceae, add a reference.

It was added.

    1.  

Figure 2. More details should be provided in then text. Explain the particular succession of sulfur bacteria depicted in Figure 2.

More details have been added in the text of the manuscript.

    1.  

Reference style is inconsistent, for example journal abbreviations.

It was checked.

Reviewer 4 Report

The paper entitled "Anoxygenic photosynthesis in photolithotrophic sulfur bacteria and their role in detoxication of hydrogen sulfide" by Dr. Kushkevych et al., is a review focused on Photolithotrophic sulfur bacteria including Chromatiaceae and Chlorobiaceae that are able to convert inorganic substrate (hydrogen sulfide and carbon dioxide) into organic matter through light energy. Due to its ability to use hydrogen sulfide, the decontamination capacity of these bacterial groups is also reviewed.

First, the figures must be improved. Figure 1 is an uninteresting (not descriptive at all figure) summary of the approach to the article. Besides being of low quality, it does not add anything to the article. Some other figures on the ultrastructure of bacterial groups for example should be incorporated.

On the other hand, I have identified several misconceptions and gramar errors that should be fixed:

  1. Some misconceptions on the text: e.g. Line 44: …….. and reduces reduced sulfur compounds as an electron donor [11].
  2. English revision needed: e.g. Paragraph lines 48-51: paragraph that presents a mess of concepts and difficult to understand.

Author Response

Dear Reviewer,

We would like to thank you for your time reviewing our manuscript. The text was carefully revised and we are thankful for your important and constructively critical comments, which improved our work. We have corrected the manuscript according to your recommendations and wrote our responses below.

Thank you for your time, positive evaluation of our work and your help.

Best wishes,

The authors

The paper entitled "Anoxygenic photosynthesis in photolithotrophic sulfur bacteria and their role in detoxication of hydrogen sulfide" by Dr. Kushkevych et al., is a review focused on Photolithotrophic sulfur bacteria including Chromatiaceae and Chlorobiaceae that are able to convert inorganic substrate (hydrogen sulfide and carbon dioxide) into organic matter through light energy. Due to its ability to use hydrogen sulfide, the decontamination capacity of these bacterial groups is also reviewed.

First, the figures must be improved. Figure 1 is an uninteresting (not descriptive at all figure) summary of the approach to the article. Besides being of low quality, it does not add anything to the article. Some other figures on the ultrastructure of bacterial groups for example should be incorporated.

This figure has been deleted from manuscript.

On the other hand, I have identified several misconceptions and gramar errors that should be fixed:

    1.  

Some misconceptions on the text: e.g. Line 44: …….. and reduces reduced sulfur compounds as an electron donor [11].

It was corrected.

    1.  

English revision needed: e.g. Paragraph lines 48-51: paragraph that presents a mess of concepts and difficult to understand.

It was corrected.

Round 2

Reviewer 2 Report

1. "....It is true that section 8 contains only one reference. This section was written based on analyzing and summarizing all literature data and include analyze and recommendation by the authors for application of anoxygenic phototrophs in hydrogen sulfide detoxication. It is not literature data where we can cite references, it is our own thoughts and recommendations to possibly set-up suh a detoxification system...."

Undoubtedly, there are literature data regarding to the issue considered in the section 8. For example, the following works (and references therein):

DOI 10.1007/978-3-319-45651-5

doi:10.3390/chemengineering3030076

Thus, the review may be improved. 

Author Response

Thank you so much for your comment and for providing reference suggestions that we overlooked. The following articles were added to the manuscript:
DOI 10.1007/978-3-319-45651-5
doi:10.3390/chemengineering3030076

Reviewer 3 Report

All my comments and suggestions have been implemented. 

Author Response

Thank you for reviewing our manuscript!

Round 3

Reviewer 2 Report

The authors improved the manuscript according to the reviewer's recommendations. Thus, the article may be accepted.